# Hyperarousal and Beyond: New Insights to the Pathophysiology of Insomnia Disorder through Functional Neuroimaging Studies

**DOI:** 10.3390/brainsci7030023

**Published:** 2017-02-23

**Authors:** Daniel B. Kay, Daniel J. Buysse

**Affiliations:** 1Department of Psychology, Brigham Young University, Provo, UT 84604, USA; 2Department of Psychiatry, University of Pittsburgh, Pittsburgh, PA 15213, USA; buyssedj@upmc.edu

**Keywords:** insomnia, neuroimaging, heuristic model, hyperarousal

## Abstract

Neuroimaging studies have produced seemingly contradictory findings in regards to the pathophysiology of insomnia. Although most study results are interpreted from the perspective of a “hyperarousal” model, the aggregate findings from neuroimaging studies suggest a more complex model is needed. We provide a review of the major findings from neuroimaging studies, then discuss them in relation to a heuristic model of sleep-wake states that involves three major factors: wake drive, sleep drive, and level of conscious awareness. We propose that insomnia involves dysregulation in these factors, resulting in subtle dysregulation of sleep-wake states throughout the 24 h light/dark cycle.

## 1. Introduction

Several models have been proposed to address the cognitive, physiological, and neurobiological features of insomnia [1]. The most prominent of these are founded on the intuitive assertion that difficulties going to sleep and staying asleep result from heightened physiological, cognitive, or emotional arousal. These types of mechanisms, which influence sleep and wakefulness, have been collectively grouped under the broad construct of “hyperarousal”. Physiological studies of patients with primary insomnia (PI) and good sleeper controls (GS) have identified peripheral nervous system (PNS) markers of hyperarousal that seem to support this intuitive explanation, linking insomnia to heightened body temperature [2], greater 24-h metabolic rate [3], and more rapid heart rate and elevated low-frequency power in heart rate, a marker of sympatho-vagal balance, during sleep [4]. Heightened cortisol levels, particularly in the pre- and early-sleep periods, have also been linked to insomnia, a finding interpreted by some as evidence of central nervous system (CNS) hyperarousal in insomnia [5,6,7,8].

Numerous studies, however, have failed to fully replicate or find any evidence of physiological hyperarousal in patients with insomnia [2,9,10,11,12,13,14,15,16,17]. “Primary insomnia” is a heterogeneous condition with respect to major symptoms (e.g., onset, maintenance, early morning awakening) and psychiatric/medical history. Thus, differences in participant samples or insomnia subtypes included in those studies may account for some of these mixed findings. Differences in methodology across studies may also play a role in these mixed findings. In particular, some of the studies that suggest a link between insomnia and physiological hyperarousal used objective polysomnography (PSG) sleep disturbances to classify insomnia [2,7]. Sleep disturbances other than insomnia (e.g., short sleep or low sleep efficiency determined by PSG, actigraphy, or self-report) are themselves associated with markers of hyperarousal [18,19,20], but are not considered diagnostic criteria for PI per se. We should also note that several studies often cited in support of the hyperarousal model of insomnia studied “poor sleepers” rather than patients diagnosed with insomnia disorder [21]. Those studies were not included in this review as they may be confounded by other sleep problems or conditions. Ultimately, mixed findings may suggest that insomnia is a heterogeneous condition with a multifaceted pathophysiology.

Studies of CNS function may provide more direct information on the nature of PI. Scalp electroencephalography (EEG), the primary measure of PSG, quantifies changes in voltage related to post-synaptic potentials, from which inferences about neuronal activity are made. Studies that use quantitative analysis of the EEG, such as power spectral analysis, suggest that patients with insomnia may experience heightened brain activation, as indexed by higher-frequency EEG power than GS, during wakefulness [22,23], the sleep onset period [22,24,25], non-rapid eye movement (NREM) sleep [26,27,28,29] or rapid eye movement (REM) sleep [28]. As with studies of the PNS, these findings have typically been interpreted within the hyperarousal framework. However, such conclusions may underestimate the complexity of normal and dysregulated sleep-wake states.

Interpretation of brain electrical activity requires careful consideration of its location, sources, and mechanisms. Critical questions to answer when interpreting EEG results include: Where does the activity occur in the brain? Is the source of the activity proximally regulated or influenced by distal brain regions or circuits? Are the mechanisms of the activity due to reduced inhibition or increased excitation? In addition, what is the influence of the activity on other brain regions? From the perspective of a hyperarousal model, one may predict that the source of higher-frequency EEG activity in PI originates from wake-promoting brain regions (ascending arousal systems) or from regional glutamatergic activity that promotes wakefulness. Answers to these questions, however, simply cannot be derived from standard PSG measures. Neuroimaging methods include a set of tools, complementary to PSG, that are beginning to more fully characterize the pathophysiology of PI. Below we review magnetic resonance spectroscopy and functional neuroimaging studies of insomnia. We introduce a heuristic model of sleep-wake states and conceptualize neuroimaging findings of insomnia within that framework. We also suggest ways that current evidence from neuroimaging studies may inform insomnia treatments.

## 2. Spectroscopy Neuroimaging Studies of Insomnia

Magnetic resonance spectroscopy (MRS) uses magnetic resonance technology and specialized pulse sequences to quantify molecular concentrations in bodily tissue, including neuromolecules in the brain. At the neuromolecular unit of analysis, neuronal activity is regulated by a complex interplay of neurochemicals, including neurotransmitters and neuromodulators. High-field MRS provides a non-invasive means to study, at the regional level, alterations in several neurotransmitter systems potentially involved in insomnia, including glutamate and γ-aminobutyric acid (GABA). Offering superior signal-to-noise than non-proton MRS, 1H (proton) MRS is commonly performed to investigate the neuromolecular alterations associated with diseases. The resonant frequency of a nucleus in a magnetic field provides information about molecular groups that carry 1H, including the types of molecules present and their relative concentrations. The temporal resolution of MRS is relatively poor, and many MRS studies do not routinely monitor inter- or intra-individual differences in EEG states during scanning. The spatial resolution of MRS is also relatively poor, with large voxel sizes (1.5–3 cm^3^) that typically span multiple brain regions. These voxels are typically positioned deep in the neocortex to avoid the edges of the brain and ventricles while still capturing important gray matter regions, including the anterior cingulate, parieto-occipital, and temporal cortices.

Glutamate is the primary excitatory neurotransmitter in the CNS. Higher glutamate in the parieto-occipital cortex has been linked to insomnia symptoms in patients with post-traumatic stress disorder [30,31] but has not been linked to PI per se [32,33,34].

The primary inhibitory neurotransmitter in the CNS, GABA is critically involved in sleep-wake regulation, including circadian sleep processes, sleep initiation, sleep maintenance, generation of slow-wave sleep oscillations, and EEG power density [35,36,37,38,39]. The role of GABA in PI remains poorly understood. Silencing GABAergic neurons and lesioning GABAergic nuclei can induce an insomnia-like state in animals [35,40]. Conversely, positive allosteric modulators of the GABA_A_ receptor (e.g., benzodiazepines, “z drugs”, and barbiturates) have sedating effects and are efficacious short-term treatments for insomnia [41,42]. With one exception that found that individuals with PI had higher GABA concentrations than GS in the occipital cortex [32], the majority of 1H MRS studies suggest insomnia is associated with lower GABA levels in the parieto-occipital cortex [30,31,33,34] or anterior cingulate [33]. Because MRS reflects presynaptic concentrations of GABA, these findings may suggest that insomnia involves impaired inhibitory control. Although both lower and higher GABA have been interpreted within the hyperarousal model of insomnia [43], the preponderance of the findings points toward lower GABA concentrations in PI that may be specific to the brain regions noted above. Such findings are consistent with the hypothesis that the higher-frequency EEG activity observed in patients with PI may be the result of reduced inhibition [44]. In other words, “hyperarousal”, as indexed by increased high-frequency EEG in PI, may be a result of impaired GABAergic inhibition rather than its cause.

## 3. Functional Neuroimaging Studies of Insomnia

Functional neuroimaging methods use a number of technologies to investigate associations between region-specific activity in the brain and specific mental functions. Common functional neuroimaging techniques that have been applied to the study of insomnia include high-density EEG, single-photon emission computed tomography (SPECT), positron emission tomography (PET), and functional magnetic resonance imaging (fMRI). Functional neuroimaging studies of healthy sleep and insomnia have been conducted during globally-defined states including active wakefulness (i.e., during various cognitive tasks), quiet wakefulness (i.e., during the resting state), NREM sleep (i.e., a quiet sleep state), and REM sleep (an active sleep state).

High-density EEG offers enhanced spatial resolution compared to standard PSG, and can identify both regional variations in brain electrical activity and the probable sources of that activity. One high-density EEG study conducted during quiet wakefulness found that individuals with PI had higher beta activity than GS in large clusters spanning the prefrontal, frontal, central, right temporal, and bilateral posterior regions of the brain during an eyes-closed resting-state condition [45]. Source localization analysis suggested that spectral EEG differences likely originated from the sensorimotor cortices [45]. However, that study found no group differences in high-frequency EEG activity during an eyes-open resting-state condition. Another high-density EEG study suggests that greater high-frequency EEG power in PI is most prominent in the left frontal and frontal midline cortices during wakefulness and sleep onset. Source localization analysis found those differences likely originated from the frontal gyri and the anterior cingulate [46]. A recent high-density EEG pilot study found that individuals with PI had greater alpha and theta levels during N3 (delta) sleep than GS, suggesting that “wake-like” brain activity persists in insomnia even during the deepest stages of sleep [47]. Source localization suggested the heighted brain activity during NREM sleep originates from sensory and sensorimotor cortical areas [47]. As noted in the previous section, increased high-frequency EEG activity during wakefulness and “wake-like” EEG activity during NREM sleep may be due to reduced inhibition as well as increased excitation in particular brain regions. Ultimately, results from high-density EEG studies are mixed and the region- and state-specific differences found in these studies cast doubts on the assertion that insomnia involves 24-h, CNS-wide hyperarousal.

Single-photon emission computed tomography (SPECT) is a technique that can be used to quantify regional cerebral perfusion (blood flow) in the brain. SPECT blood flow studies involve the injection of a gamma-emitting tracer (e.g., technetium (^99m^Tc) exametazime) into the bloodstream and subsequent detection using a gamma camera. A SPECT study in patients with depression linked insomnia symptom severity to hypoperfusion in the right subgenual and rostral anterior cingulate cortex, left claustrum, and left insula during quiet wakefulness [48]. Hypoperfusion has also been documented in patients with PI compared to GS in the basal ganglia, medial frontal, occipital, and parietal regions during NREM sleep [49]. A reasonable assumption of a hyperarousal model of insomnia is that blood flow would be positively associated with “arousal”, at least in brain regions associated with arousal. Such an assumption is difficult to prove, and some have argued that lower blood flow may be consistent with a hyperarousal model of insomnia [50]. Although it is unlikely that gross blood flow is a sensitive correlate of arousal, lower blood flow during wake and NREM sleep seems counterintuitive from the perspective of a general hyperarousal model of insomnia.

Positron emission tomography (PET) neuroimaging involves the injection of a positron-emitting radioactive tracer into the bloodstream, with subsequent distribution to all tissues including the brain. Specific tracers can be used to measure the density of neurotransmitter receptors or to measure regional brain metabolic rate during stable global brain states. [^18^F]Fluorodeoxyglucose positron emission tomography (FDG-PET) can provide a snapshot of regional glucose metabolism that occurs in the brain over a short period of time (e.g., 20 min) during which the radioactive tracer circulates and is taken up by tissues in proportion to local metabolic activity. A preliminary FDG-PET study in a small sample of patients with PI (*n* = 7) and GS (*n* = 12) reported three major findings. First, patients with PI had smaller sleep-wake differences in relative regional cerebral metabolic rate for glucose (rCMR_glc_) than GS in the medial prefrontal, anterior cingulate, insula, thalamus, hippocampus, amygdala, hypothalamus, and brainstem. Second, during the wake state, individuals with PI had lower relative rCMR_glc_ than GS in a broad region of the frontal cortex bilaterally, left superior temporal gyrus, left occipital cortex, left parietal cortex, thalamus, hypothalamus, and brainstem reticular formation. Third, individuals with PI had greater whole-brain glucose metabolism than GS across sleep-wake states [51]. The results of that study have been widely cited as support for the hyperarousal model of insomnia.

A more recent FDG-PET study in a larger sample of individuals with PI and GS from the same laboratory resulted in three major observations. First, individuals with PI had a smaller sleep-wake difference than GS in relative rCMR_glc_ in brain regions involved in cognition, self-referential processes, and affect. These regions include the left frontoparietal, occipital, lingual/fusiform, and precuneus/posterior cingulate cortices. Second, during quiet wakefulness, individuals with PI had lower relative rCMR_glc_ than GS in widespread regions of the brain spanning the neocortex to the brainstem (Figure 1). Patients with PI had higher relative rCMR_glc_ than GS only in the cerebellum during the waking state. During NREM sleep, individuals with PI had lower relative rCMR_glc_ than GS in clusters limited to limbic brain regions, many of of which are involved in the salience and default mode networks, including the anterior cingulate, medial frontal gyrus, orbitofrontal cortex, inferior frontal gyrus, right posterior cingulate, and bilateral precuneus (Figure 2). See Table 1 for a complete list of brain regions showing group differences in relative rCMR_glc_ during wake and NREM sleep in PI compared to GS. Third, no group differences were identified in semi-quantitative whole-brain glucose metabolism [52]. Although smaller sleep-wake differences can be explained within a hyperarousal framework, lower relative rCMR_glc_ in patients with PI compared with GS during NREM sleep in limbic brain regions is more difficult to explain within this framework. Collectively, these results highlight the complex regional pathophysiology of insomnia across sleep-wake states, a complexity that is not easily explained by a global hyperarousal mechanism.

Functional magnetic resonance imaging (fMRI) most commonly measures blood-oxygen-level dependent (BOLD) contrasts to detect changes in the magnetic field gradient that are associated with blood volume, oxygenation, and flow. The underlying assumption is that activation of a brain region results in increased utilization of oxygen and an even greater increase in blood flow, with a net increase in the BOLD signal. Modeling BOLD fluctuations over time or in response to a stimulus allows for inferences about task-based regional activity, resting-state regional activity, and resting-state functional connectivity.

Task-based fMRI. Task-based fMRI models regional brain changes in the BOLD signal in response to stimuli. One task-based fMRI study found that patients with PI had lower activity in the left medial and inferior frontal gyrus than GS during executive control paradigms (i.e., letter and category fluency) [54]. Another task-based fMRI also found that patients with PI had widespread hypoactivation during an executive functioning paradigm (i.e., n-back) [55]. Compared to GS, PI patients had lower activity during the n-back task in the frontal cortex, left cingulate, premotor area, left supplementary motor area, thalamus, parietal lobe, left posterior cingulate, precuneus, left cuneus, and right cerebellum [55]. During another executive functioning fMRI paradigm, a spatial working memory task, PI patients showed lower activity in the parahippocampal gyri, temporal cortex, frontal cortex, and superior parietal lobule and higher activity than GS in the left temporal, left occipital, and right frontal cortices [56]. Another task-based fMRI study presented participants with arousing stimuli and did not find any differences between individuals with PI and GS in amygdala activation to emotionally arousing stimuli [57]. Furthermore, PI patients actually had lower amygdala activation to non-insomnia-related stimuli [57]. Patients in that study did, however, have greater amygdala reactivity to stimuli with insomnia-related content, suggesting that the association between insomnia and heightened arousal may involve a stimulus-specific or top-down cognitive process [57]. These task-based fMRI studies generally point to hypoactivation during executive control tasks and response to non-insomnia-related stimuli, normal activation to emotionally arousing stimuli, and hyperactivation in response to insomnia-related stimuli.

Resting-state brain activity. Methods used to assess differences in brain activity during the fMRI resting-state scan include regional homogeneity (ReHo) and amplitude of low-frequency fluctuations (ALFF). Regional homogeneity is used to quantify the extent to which voxels are synchronized, across a time series, with their nearest neighbors within a specific cluster [58]. Higher regional ReHo values may represent greater intrinsic brain activity. Amplitude of low-frequency fluctuations is used to quantify the regional amplitude of slow fluctuations in spontaneous brain activity [59]. Higher or lower ALFF values are interpreted as regional abnormalities in spontaneous BOLD signals. These fMRI methods are relatively new but have been used to identify altered regional brain activity in various patient samples [59,60], including insomnia.

Two studies used ReHo to assess group differences (PI vs. GS) in brain activity during an fMRI resting-state scan [61,62]. One of these studies found that, compared to GS, patients with PI had lower homogeneity in the cingulate cortex and right cerebellum and higher regional homogeneity in the left fusiform gyrus [61]. A later study also found both higher and lower homogeneity in patients with PI compared to GS [62]. Specifically, compared to GS, patients with PI had lower homogeneity in the right middle cingulate and left fusiform gyrus and higher homogeneity in the left insula, right anterior cingulate, bilateral precentral gyrus, and left cuneus [62].

Two studies used ALFF to assess group differences (PI vs. GS) in brain activity during an fMRI resting-state scan. One study found that patients with PI had lower ALFF than GS in the orbital gyrus, right rectal gyrus and right cerebellum and higher ALFF than GS in the posterior cingulate, right lingual gyrus, left precuneus, and left cuneus [63]. Another study in healthy participants found that, compared to GS, individuals with insomnia symptoms had lower fractional ALFF (i.e., ALFF relative to the whole frequency range) in the left ventral anterior insula, bilateral posterior insula, left thalamus, and pons and higher fractional ALFF in the middle occipital gyrus and right precentral gyrus [64].

Although there is some overlap in the brain regions identified as being altered in PI during the resting state, the direction of results was not consistent across these studies. A major limitation of these resting-state fMRI studies is the lack of EEG monitoring to confirm patients’ states. About one third of participants drifts into sleep within 3 min of starting a resting state scan [65]. Thus, individual differences in states during the scans may contribute to the contradictory findings. Alternatively, these mixed findings may suggest insomnia involves dysregulated brain functioning, most robustly in the cingulate cortex, left insula, left cuneus, and left fusiform during quiet wakefulness.

Resting-state functional connectivity. Functional connectivity quantifies regional patterns of synchronous brain activity during a resting-state fMRI scan while an individual is not performing any particular task [66]. Functional connectivity is inferred based on spontaneous fluctuations in the BOLD signal over time. Collinearity of the BOLD signal between spatially distinct brain regions is thought to reflect greater functional connectivity of those regions. This approach has been used to identify several resting-state networks (e.g., salience, default mode, executive control), that is, spatially distinct brain regions that share collinear patterns of increased and decreased activity over time [67].

Several fMRI studies have sought to test whether individuals with PI have altered functional connectivity in the salience network. The salience network consists of limbic brain regions, including the amygdala, insula, and anterior cingulate, that are involved in reactivity to novel, important, or emotional stimuli [68,69]. One study found no connectivity differences between PI and GS in the salience network during resting state, but higher connectivity of the anterior insula with the anterior salience network in PI when participants were instructed to fall asleep [70]. That result may suggest that altered functional connectivity in the salience network is perhaps situation-specific (i.e., trying to fall asleep) in PI. Another study found that, compared to GS, patients with PI had lower amygdala connectivity with the right inferior frontal gyrus, bilateral superior temporal gyrus, insula, left thalamus, bilateral caudate, bilateral lentiform nucleus, and left lateral globus pallidus [71]. Thus, that study found lower connectivity among major nodes of the salience network, a finding that is consistent with a data-driven connectivity study that found PI had lower connectivity than GS among major nodes of the salience network including the right insula with the anterior cingulate and with the left insula [72]. These two studies also found PI had higher connectivity involving sensorimotor regions of the brain, a finding that builds on a previous connectivity study in healthy individuals that found insomnia symptoms were associated with greater connectivity in primary sensory/motor regions of the brain [73]. Although higher connectivity of sensorimotor regions of the brain may be explained by hyperarousal mechanisms, they may also be explained by impaired inhibitory processes. Regardless, findings of lower connectivity within major nodes of the salience network during resting state are not consistent with the view that insomnia involves 24-h, emotion-related hyperarousal.

Other resting-state functional connectivity studies have sought to determine whether patients with PI have connectivity alterations in the default mode or executive control networks. Major nodes of the default mode network include the posterior cingulate cortex and adjacent precuneus, inferior parietal lobules, medial temporal lobe, and medial frontal cortices [74]. One study found that individuals with PI had lower functional connectivity within the default mode network in specific areas including the medial prefrontal cortex with the right medial temporal lobe and the left medial temporal lobe with the left inferior parietal cortices [75]. Another study that used the superior parietal lobule (a major node of both executive control and default mode networks) as the seed region found that patients with PI, compared to GS, had higher connectivity of this region with the anterior cingulate, left posterior cingulate, right inferior frontal gyrus, right insula, and precuneus [76]. These are all regions of the brain involved in conscious awareness. That study also found lower connectivity of the superior parietal lobule with the right dorsolateral prefrontal cortex, a major node of the right executive control network [76]. These findings overlap somewhat with a data-driven study that found insomnia involves lower functional connectivity between major nodes of the default mode networks (e.g., the right inferior parietal cortex and the posterior cingulate) and lower functional connectivity involving major nodes of the executive control network (i.e., the prefrontal cortex with the middle cingulate and insula) [72].

## 4. Summary of Neuroimaging Studies of Insomnia

Neuroimaging studies are beginning to broaden our understanding of the pathophysiology of insomnia beyond the hyperarousal model. Although some counterevidence has been reported [77], several EEG studies suggest insomnia is associated with greater high-frequency EEG activity across sleep-wake states that appears to be regionalized most strongly to the frontal lobes or sensorimotor regions. Seemingly contradictory are functional neuroimaging studies that suggest insomnia is associated with reduced brain activation, blood flow, or glucose metabolism during active wake, quiet wake, and NREM sleep in major nodes of the executive control, salience, or default mode networks. The majority of MRS studies suggest that insomnia is associated with reduced regional GABA concentrations in the medial parieto-occipital cortex and anterior cingulate. Based on this evidence, we propose that higher-frequency EEG activity in the presence of regionalized hypoactivation/hypoperfusion/hypometabolism may be explained by impaired inhibitory processes in the brain.

Resting-state fMRI studies have produced seemingly mixed findings as well; all suggest insomnia involves both increased and decreased activity, most consistently in the cingulate cortex, left insula, left cuneus, and fusiform. Resting-state fMRI connectivity analyses suggest insomnia is associated with lower connectivity in the salience network, lower connectivity in the executive control network, higher connectivity in sensorimotor regions, and altered connectivity in the default mode network. It is possible these mixed findings suggest that insomnia is a heterogeneous condition with a multifaceted pathophysiology. It is also possible that insomnia involves altered functioning in several brain regions that play a role in conscious awareness, executive control, salience, and sensory processing during quiet wakefulness but that these manifest differently in different patients.

Collectively, the results from neuroimaging studies have been interpreted in the literature as providing support for a hyperarousal mechanism of insomnia regardless of the brain regions involved, the neuromolecules being studied, or the direction of the results. Findings that those with PI, compared to GS, had normal amygdala reactivity to emotional stimuli, and hypoactivation/ hypoperfusion/hypometabolism in many brain regions including major nodes of the salience network during quiet wake or NREM sleep, and consistently lower resting-state functional connectivity in the salience network, do not support the view that insomnia involves a 24-h, CNS-wide or emotion-related hyperarousal. Brain regions identified as having greater activity/connectivity tend to be specific to insomnia-related stimuli or circumstances (trying to fall asleep) and cluster most strongly in brain regions involved in conscious awareness. We propose that neuroimaging studies suggest the pathophysiology of insomnia is more complex than can be explained by a global hyperarousal mechanism.

## 5. Heuristic Model of Sleep-Wake States

We propose that sleep-wake states are determined by the combination of three major factors: wake drive, sleep drive, and level of conscious awareness (Figure 3). The factors within the model, as described below, were derived conceptually. Previously, Kay and colleagues applied this constructivist model to the understanding of insomnia in older adults [78]. Herein, we explore the results of functional neuroimaging studies of insomnia within this model.

The wake drive factor may map onto the “hyperarousal” mechanism proposed by several models of insomnia and includes global brain processes that are regulated by multiple nuclei in the brainstem, midbrain, and basal forebrain, including the laterodorsal hypothalamus, tuberomammillary nucleus, pedunculopontine nucleus, lateral tegmental nucleus, substantia nigra, ventral tegmental area of the midbrain, basal and medial hypothalamus, locus coeruleus, and raphé nuclei [79]. These systems may be engaged to promote wakefulness through bottom-up stimulus-driven mechanisms, intrinsic processes (e.g., circadian rhythms), and top-down cognitive processes that involve activation/disinhibition of the task-positive network: dorsolateral and ventrolateral prefrontal regions, insular cortex, supplementary motor area (SMA) and the pre-SMA [80]. We note here that activation/disinhibition of cortical brain regions is not itself wake-promoting nor is it necessarily a marker that wake-promoting systems are activated. Rather, wake drive manifests in the cortex when activated by systems in the brainstem, midbrain, and basal forebrain. These systems are engaged during executive functioning paradigms and may also be activated by the administration of drugs such as caffeine and amphetamines that promote wakefulness.

Sleep drive is the second major factor of this model. In other models, sleep drive is often conceptualized as falling on the opposite end of the same spectrum as wake drive. Although healthy wake states typically involve low sleep drive, these factors are regulated by different circuitry and can function independently from one another [81]. For example, co-existing high sleep drive and high wake drive is a hallmark of sleep deprivation. Sleep drive includes both global sleep promoting networks, including the preoptic areas of the hypothalamus and thalamus, and regional use-dependent sleep processes [79,82]. Higher sleep drive is associated with higher-amplitude, lower-frequency EEG brain wave patterns in the frontal cortex during wake and NREM sleep reviewed in [83] and lower cerebral metabolism in the orbitofrontal cortex and thalamus during NREM sleep [84].

Level of conscious awareness is the third major factor in this model. Loss of conscious awareness and sleep are often conceptualized as being synonymous. Although healthy sleep states typically involve perceived loss of conscious awareness, many individuals experience a conscious state during EEG-defined sleep. Indeed, 50% of GS report being awake following the first sleep spindle and 5% report being awake following the onset of delta sleep [85,86]. Conscious awareness during NREM and REM sleep is reduced but some level of conscious awareness persists, as is evidenced by dream mentation [87]. Moreover, loss of or reduced conscious awareness does not constitute sleep. Thus, conscious awareness falls on a continuum from high to low across sleep-wake states. Conscious awareness, as reported by participants, relates to activation of the insula and the adjacent claustrum, anterior and posterior cingulate cortices, precuneus, and the left prefrontal cortex [88,89,90,91,92]. Higher levels of conscious awareness are also associated with higher fMRI-assessed functional connectivity of the default mode and executive control networks [93,94]. Reduced conscious awareness during NREM sleep is also associated with reduced EEG-assessed effective connectivity of the cortex [95]. Our discussion of levels of conscious awareness in the heuristic model focuses on the self-report unit of analysis, as it relates to perception of sleep-wake states and the presence of mental content that can be retrieved upon awakening. Nevertheless, future research at other units of analysis (i.e., circuits, physiology, and behavior) may help elaborate on this factor in relation to markers of altered exteroceptive or interoceptive levels of conscious awareness in insomnia.

In this model, each of these aforementioned factors is conceptualized as varying along its own continuum. Although these factors work in harmony in healthy individuals, we will consider them as orthogonal for heuristic purposes. Global states are constructed based on where an individual falls within these factors at any given moment. Although there are theoretically an infinite number of potential global states, we present a simplified diagram involving eight major sectors in Figure 3. The diagram includes four sectors where states typically occur in healthy individuals: (1) active wake; (2) quiet wake; (3) REM sleep; and (4) NREM sleep. Individuals with PI may experience altered states that shift toward the other four sectors: (5) active wake with reduced conscious awareness; (6) REM sleep with heightened conscious awareness; (7) quiet wake with reduced conscious awareness; and (8) NREM sleep with heightened conscious awareness.

Healthy active and quiet wake. During wakefulness, healthy states oscillate during the day between active wakefulness (sector 1, where wake drive is high, sleep drive is low, and conscious awareness is high) and quiet wakefulness (sector 2, where wake drive is reduced but sleep drive remains low and conscious awareness remains high). The healthy active wake state during task-engagement is associated with activation of the arousal systems, inhibition of the default mode network [96], and engagement of the task-positive network including frontoparietal cortices, insula, and supplemental motor areas [80].

The level of wake drive is the primary difference between active and quiet wake. Relative to active wake, the healthy quiet wake state is associated with reduced activity in the arousal systems, inhibition of the task-positive network, and activation of the default mode network. Conscious awareness remains high and sleep drive remains low during quiet wakefulness.

Healthy NREM and REM sleep. Healthy sleep states oscillate between quiet (NREM) and active (REM) sleep states. During healthy NREM sleep, whole-brain glucose metabolism decreases relative to wake [52]. The greatest reductions occur in heteromodal regions of the frontal and parietal cortices, posterior cingulate cortex, and thalamus [52,97]. Thus, the NREM sleep state is most consistent with sector 4 where sleep drive is high, wake drive is low, and conscious awareness is low.

Healthy REM sleep, relative to NREM sleep, is characterized by increased activity in the brainstem, basal ganglia, basal forebrain, anterior inferior insula, hippocampus, anterior cingulate, medial prefrontal, and visual/auditory association cortices [84,98]. We propose that this paradoxical state involves activation of circuitry associated with the wake drive in the presence of high sleep drive, as reflected by relatively lower blood flow or metabolic rate in the posterior insula, posterior cingulate, and frontal heteromodal regions of the frontal and parietal cortices [84,98]. Healthy REM sleep may also involve somewhat increased conscious awareness (i.e., interoceptive awareness) due to activation of the anterior cingulate and insula but not the levels experienced during healthy wakefulness. In this model, the healthy REM sleep state is most consistent with sector 3, where sleep drive is high, wake drive is heightened, and conscious awareness remains relatively low.

## 6. Insomnia

The combination of the three major factors in the heuristic model can be applied to the myriad states associated with many forms of medical and psychiatric conditions. Below, we discuss state- and region-specific differences associated with insomnia in the context of the above heuristic model.

Active wake. While engaged in executive control tasks, individuals with PI demonstrate hypoactivation in several nodes of the executive control network, including the left medial prefrontal cortex and left inferior frontal gyrus, compared to GS [54,55,56]. Despite these functional neuroimaging differences, in two of these studies the patients with PI performed comparably to or better than GS on executive tasks, suggesting that compensatory mechanisms may have been employed [54,55]. This may suggest that wake-promoting mechanisms are intact in these patients. This pattern may be most consistent with an altered state as characterized by sector 5 in our model, where sleep drive is low and wake drive is high but conscious awareness is attenuated. Because subsequent sleep depth in these brain regions is influenced by daytime use, interventions that help engage executive networks during active wakefulness in PI may prove to be effective for many patients. Indeed, one study found that an eight-week, home-based, personalized, computerized cognitive training program that engages high-level cognitive functioning may improve insomnia symptoms [99].

Quiet wake. Higher and lower regional brain activity and higher and lower functional connectivity have been reported in PI compared to GS across and within neuroimaging studies. These findings point to a more complex pathophysiology of insomnia than can be explained by increased global wake drive. High-density EEG studies suggest that insomnia may be associated with heightened region-specific brain activity during quite wake, but the specific brain regions involved remain unclear [45,46]. Altered activity and functional connectivity during quiet wake occurs most consistently in primary sensory-motor regions, frontoparietal regions, and nodes of the default mode network [52,61,63,64,75]. This pattern may be explained by dysfunction of inhibitory processes in the cortex. Less inhibition of the task-positive network may result in greater inhibition of the default mode network, thus explaining the presence of heightened brain activity in major nodes of the executive control network in some studies. Thus, rather than originating from heightened wake drive, these findings may be due to alterations in cortical inhibitory or conscious awareness mechanisms during quiet wake. These patterns are most consistent with an altered state of wakefulness consistent with sector 7, where wake drive is low but the task-positive network is disinhibited, sleep drive is low, and conscious awareness associated with the default mode network is inhibited (i.e., low). Interventions that engage the default mode network in PI may prove to be useful as this may increase conscious awareness during wakefulness and inhibit the task-positive network during resting states. There is, indeed, some evidence that mindfulness meditation, which engages the default mode network, improves insomnia symptoms [100]. Conversely, inhibition of the task-positive network (e.g., via repetitive transcranial magnetic stimulation of the dorsolateral prefrontal cortex) may also be a means of targeting the imbalance in functional brain activity associated with insomnia during quiet wake [101].

NREM sleep. Compared to GS, NREM sleep in PI is associated with lower blood flow, particularly in the basal ganglia, frontal medial, occipital, and parietal regions during NREM sleep [49]. A more recent study demonstrated that insomnia was associated with lower relative regional glucose metabolism in several limbic brain regions during NREM sleep [52]. These patterns may reflect heightened sleep drive during PSG-defined NREM sleep in those regions. The latter study also found smaller sleep-wake differences in relative glucose metabolism in the left frontoparietal and precuneus/posterior cingulate, which suggests less reduction of brain activity in regions of the brain involved in higher-order conscious awareness [52]. Collectively, NREM sleep in PI is most consistent with sector 8, where sleep drive is high, wake drive is low, and conscious awareness is high. Thus, treatments that facilitate a reduction in activity in higher-order consciousness centers of the brain (left frontoparietal and posterior cingulate) during NREM sleep may prove to be effective. This may be the mechanism of effect for hypnotic sleep aids, however, more precise drug delivery to consciousness centers of the brain may improve pharmaceutical treatments for insomnia.

REM sleep. Little is known about regional brain activity during REM sleep in PI. Heightened high-frequency EEG activity during REM sleep has been identified in PI [28]. However, a more recent EEG study found that insomnia was associated with lower beta activity in the prefrontal cortex during REM sleep [102]. Interestingly, a PSG study found that sleep discrepancy for wake time after sleep onset, that is, reporting greater wake time than was measured with PSG, occurred during REM sleep [103]. One possibility is that insomnia involves even higher sleep drive in the presence of greater conscious awareness during REM sleep, consistent with sector 6 in our model. Neuroimaging studies of sleep during REM sleep in PI are needed to test this possibility.

## 7. Treatment

Cognitive behavioral therapy for insomnia (CBTI) is recommended as the first-line treatment for chronic insomnia [104,105]. The mechanisms through which CBTI improves insomnia remain unclear. The few neuroimaging studies conducted on CBTI suggest that this treatment improves several of the neural alterations that differentiate insomnia patients from GS. Despite not being designed to specifically target brain regions associated with insomnia, techniques used in CBTI have been shown to increase brain activity during engagement in tasks of executive functioning(i.e., letter and category fluency) [54]; and increase regional blood flow during NREM sleep [106] in brain regions where they were lower than in GS at baseline. Treatment components of CBTI are often conceptualized in the literature as reducing arousal by enhancing parasympathetic activation (e.g., relaxation), removing arousing stimuli from the sleep environment (i.e., stimulus control), and aligning the sleep drive with the circadian phase when it has the lowest intrinsic wake drive (e.g., sleep restriction). However, evidence for reduced hyperarousal as a mechanism of CBTI is scant. One study found that body temperature decreased following CBTI [107]. In addition, contrary to the general hyperarousal hypothesis, low frequency heart rate variability [108] and cortisol [107] tend to increase following CBTI. Thus, CBTI may work on different mechanisms than reducing wake drive during the desired sleep time. The active ingredients in CBTI, stimulus control and sleep restriction, limit the amount of time patients spend in bed to little more than is needed to obtain adequate sleep. Because both approaches result in decreased total sleep time, they appear to primarily promote sleep drive in these patients. In terms of conscious awareness, there is evidence that CBTI reduces subjective-objective sleep discrepancy following treatment [109]. It is possible that CBTI works to suppress conscious awareness during sleep by enhancing sleep drive, thereby reducing sleep discrepancy.

We propose that conceptualizing insomnia symptoms within the heuristic model described above may help guide treatment for insomnia. The most important consideration is that insomnia is a state- and brain region-specific sleep disorder. Because daytime brain activity and sleep patterns are related [110], it may be possible to hone treatment components to address brain alterations in insomnia respective to the state. For example, increasing activation of left frontoparietal network during task engagement and the default mode networks during quiet wakefulness may, via use-dependent sleep processes, facilitate reductions in conscious awareness during NREM and REM sleep. Likewise, reducing conscious awareness during sleep in PI may result in greater restoration of those brain regions, thereby contributing to greater daytime functioning in those regions during wakefulness. Mindfulness meditation, repetitive transcranial magnetic stimulation, and cognitive training have shown promise for treating insomnia [99,100,101] and may help individuals engage inhibitory processes during wakefulness, thereby enhancing use-dependent sleep drive in affected regions for subsequent sleep. The second consideration is that the model proposed is a heuristic and there is likely variation in the combination and degree to which each of the major factors contribute to an individual patient’s symptoms. Thus, this model may help clinicians conceptualize the heterogeneity of symptoms present across different patients with insomnia. Moreover, the presentation of symptoms of each patient can be applied to this model to help guide treatment. For example, patients presenting with more acute insomnia, insomnia with short sleep duration, more severe insomnia symptoms, or insomnia associated with stress, may have greater wake drive. These patients would benefit from treatment elements that reduce arousal (e.g., stimulus control, relaxation, and scheduled worry techniques) or target brain circuitry associated with the wake drive. Other patients may present with more highly erratic night-to-night patterns of sleep, involving variable patterns of sleep deprivation and sleep rebound. Because dysfunction in sleep processes may be involved in maintaining these patients’ insomnia symptoms, treatments that enhance and stabilize the sleep drive such as sleep restriction and regular wake up times imposed as part of stimulus control may prove most effective for these patients. Ultimately, this model can be used to develop individualized treatment plans to target the symptoms and potential brain region- and state-specific mechanisms involved in their insomnia symptoms.

## 8. Conclusions

Functional neuroimaging studies suggest insomnia is associated with state- and region-specific alterations in brain functioning. We propose a heuristic model of sleep-wake states for understanding the pathophysiology of insomnia. This model involves three major factors—wake drive, sleep drive, and conscious awareness—that work together to produce the major sleep-wake states experienced across the 24-h day. The brain regions associated with altered sleep-wake differences in insomnia cluster in brain regions whose activation/deactivation is associated with levels of conscious awareness (i.e., left frontoparietal cortex and precuneus/posterior cingulate). These findings conflict with physiological or neurobiological explanations of the pathophysiology of insomnia that treat sleep and wakefulness as uniform, global phenomena. This model may be used as a heuristic for understanding the heterogeneous set of symptoms experienced by patients with insomnia across sleep-wake states. Larger scale neuroimaging studies that capture the heterogeneity of insomnia can further elucidate the pathophysiology of insomnia. Additional work on how brain alterations across sleep-wake states relate to daytime impairments experienced by patients with insomnia is also needed. In particular, neuroimaging studies may help define the mechanisms through which insomnia confers risk for daytime impairments and depression.

In conclusion, we echo the sentiments of Professor C.V. Economo [111] when he declared, “we are ardently seeking to find methods to excite externally either by electricity or rays or diathermy through the skull, the centers of our nervous system in the intention of producing a therapeutic effect. Some initial results have already been obtained…Imagine we once had an effective method of influencing deep lying centers, in this case the exact knowledge of the localization of the center for sleep regulation … would make it possible to treat insomnia and other sleep disturbances in a better and more active way than by drugs or by the roundabout way of hydrotherapy and psychotherapy. Let us express the hope that we shall soon be able to have such results” (p. 259).

## Figures and Tables

**Figure 1 brainsci-07-00023-f001:**
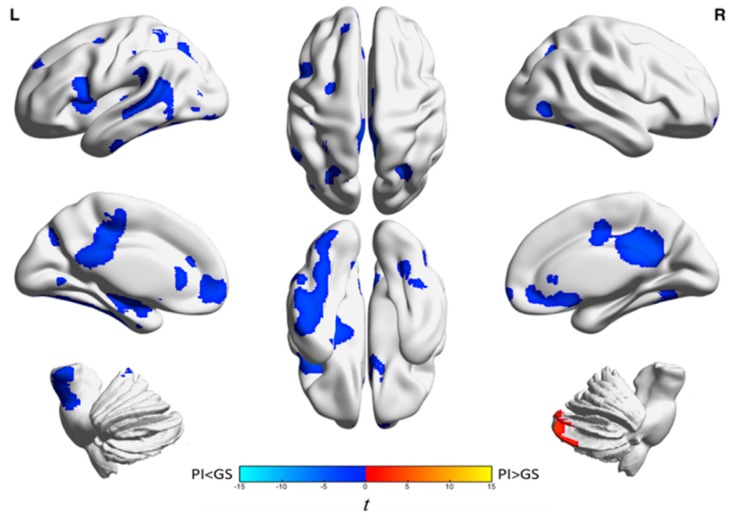
Group differences in relative glucose metabolism during wakefulness. We assessed relative regional cerebral metabolic rate for glucose (rCMR_glc_) in a sample of 44 patients with primary insomnia (PI) and 40 good sleeper controls (GS) during morning wakefulness. Patients with PI had lower relative rCMR_glc_ in four clusters spanning the neocortex and brainstem. Patients with PI also had higher relative rCMR_glc_ than GS in the right cerebellum. All clusters were significant at *p*_3DC_corrected_ < 0.05. A full list of brain regions involving these clusters is presented in Table 1. The color bar represents t values; blue indicates regions where patients with PI had lower relative rCMR_glc_ than GS and orange indicates regions where patients with PI had higher relative rCMR_glc_ than GS during wakefulness. This figure was originally published in the journal *Sleep* [52]. Used with permission. Note: L indicates the left side of the brain, R indicates the right side of the brain, and 3DC_corrected indicates that familywise error (FWE) correction and clusterwise extent thresholds were determined using 3dClustSim [53].

**Figure 2 brainsci-07-00023-f002:**
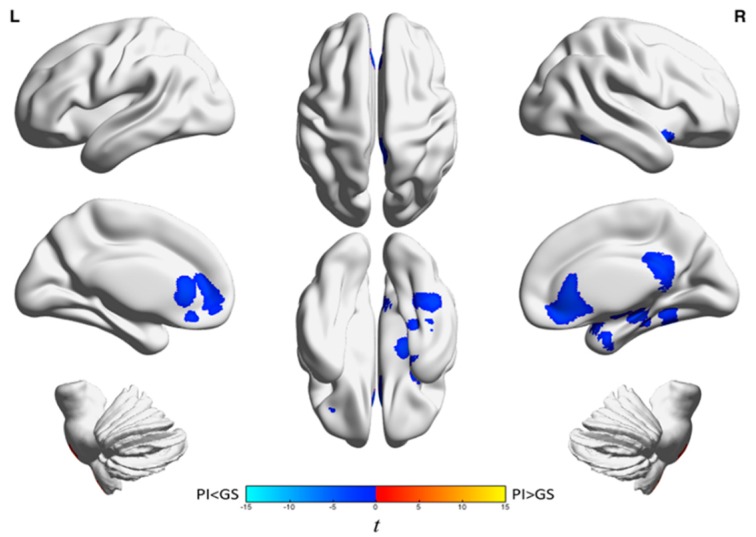
Group differences in relative glucose metabolism during non-rapid eye movement (NREM) sleep. We assessed relative regional cerebral metabolic rate for glucose (rCMR_glc_) in a sample of 44 patients with primary insomnia (PI) and 40 good sleeper controls (GS) during NREM sleep. Patients with PI had lower relative rCMR_glc_ in three clusters centered on the anterior cingulate, right medial temporal lobe, and right precuneus/posterior cingulate; *p*_3DC_corrected_ < 0.05 for all clusters. A full list of brain regions involving these clusters is presented in Table 1. The color bar represents t values; blue indicates regions where PI had lower relative rCMR_glc_ than GS during NREM sleep. This figure was originally published in the journal *Sleep* in 2016 [52]. Used with permission. Note: L indicates the left side of the brain, R indicates the right side of the brain, and 3DC_corrected indicates that familywise error (FWE) correction and clusterwise extent thresholds were determined using 3dClustSim [53].

**Figure 3 brainsci-07-00023-f003:**
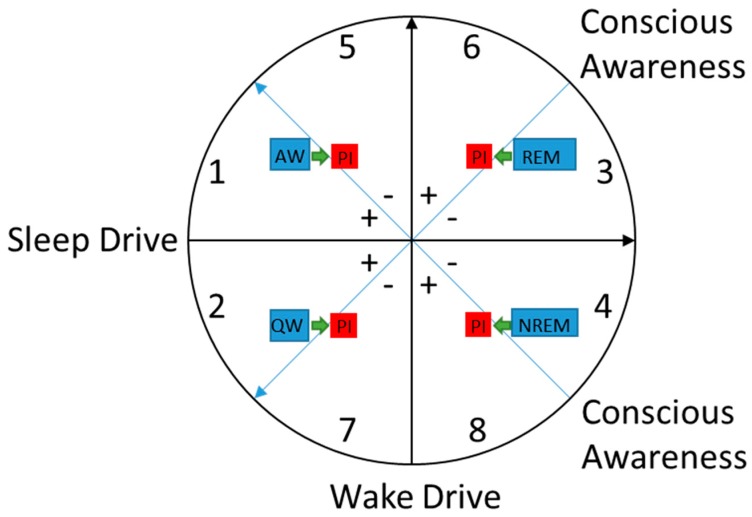
A heuristic model of sleep-wake states. In this model global states are represented in 2-dimensional space based on three major factors: sleep drive, wake drive, and level of conscious awareness. The black arrow pointing toward the top of the circle represents greater wake drive. The black arrow pointing toward the right of the circle represents greater sleep drive. The two blue-lined arrows pointing to the left side of the circle represent higher conscious awareness. To help visualize the conscious awareness factor in relation to the other two factors in the model, the + symbols represent sectors where conscious awareness is higher and the − symbols represent sectors where conscious awareness is lower. Healthy states are indicated by the blue boxes. Healthy active wakefulness (AW) occurs in sector 1 and healthy quiet wakefulness (QW) occurs in sector 2. Healthy rapid-eye movement (REM) sleep occurs in sectors 3 and healthy non-rapid eye-movement (NREM) sleep occurs in sector 4. States experienced by patients with primary insomnia (PI) are indicated by the red boxes. The green arrows represent the hypothesized state shifts experienced by patients with PI relative to good sleepers. Patients with PI may have altered states across sleep-wake states due to reduced activation in brain regions involved in conscious awareness during the wake states and heightened brain activation in these regions during sleep.

**Table 1 brainsci-07-00023-t001:** Group (PI vs. GS) differences in relative glucose metabolism during wakefulness and NREM sleep.

Analysis	Brain Region	*k* ^A^	*t*-Statistic (Max) ^B^	*x*	*y*	*z*
Wake	Left frontal cortex and anterior cingulate gyrus	1439	−4.0	−18	38	−4
	Left inferior frontal gyrus and left insula	947	−44	−44	14	14
	Right medial frontal gyrus, anterior cingulate, frontal-orbital gyrus, superior frontal gyrus, and caudate	1128	−4.7	14	28	−10
	Temporal lobe, parietal lobe, precuneus, middle and posterior cingulate gyri, frontal lobe, occipital lobe, left hippocampus, putamen, insula, left brainstem, and left amygdala	11,925	−5.4	26	−60	−36
	Right cerebellum	729	3.8	16	−88	−32
NREM	Anterior cingulate, medial frontal gyrus, orbitofrontal cortex, inferior frontal gyrus, and right caudate	2335	−4.6	14	30	−10
	Right posterior cingulate, bilateral precuneus, and middle cingulum	1100	−5.3	12	−40	20
	Right fusiform gyrus, parahippocampus, superior and inferior temporal gyri, hippocampus, and amygdala	2076	−5.1	38	2	−24

Note: ^A^ Cluster sizes (*k*) greater than 670 voxels for wake and 708 voxels for NREM sleep were significant at height threshold *p_uncorrected_* > 0.005 and cluster threshold *p_3DC_corrected_* < 0.05; ^B^ Negative *t*-statistics indicate regions where patients with PI had lower relative rCMR_glc_ than GS; positive *t*-statistics indicate regions where patients with PI had higher relative rCMR_glc_ than GS. This table was originally published in the journal *Sleep* in 2016 [52]. Used with permission. Note: PI indicates primary insomnia, GS indicates good sleeper controls, NREM indicates non-rapid eye movement sleep, columns labeled as *x*, *y*, *z* indicate brain coordinate, and rCMR_glc_ indicates relative regional cerebral metabolic rate for glucose.

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
