# Peer review of "Hyperarousal and Beyond: New Insights to the Pathophysiology of Insomnia Disorder through Functional Neuroimaging Studies"

_brainsci, 2017, doi:10.3390/brainsci7030023_

Round 1

Reviewer 1 Report

General comments

The manuscript by Kay and Buysse reviews the insights provided by functional neuroimaging into the pathophysiology of insomnia, and applies a heuristic model of sleep-wake states to the specific case of insomnia. This is a very interesting topic, and an up to date review is needed. The manuscript itself it well written and generally clear, and makes a very good case for conceptualising insomnia as a complex disorder characterised by state- and region-dependent abnormalities in brain function. I have only a few relatively minor comments.

Specific comments

Page 3 line 103: are these examples of higher GABA concentrations in good sleepers observed during wakefulness? Are there studies MRS studies during sleep in GS or PI?

Page 3 line 110: in keeping with the general view expressed in the manuscript that PI is a complex, multifaceted disorder, I would be surprised if GABA was globally lower in PI compared to GS – would it not be more likely that there would be regionally-specific variations?

Page 3 line 122: typo, REM is an active sleep state

Page 3 line 128: source localisation is a more common term that source generation?

Page 3 line 144: SPECT can be used with other tracers than 99mTc and other physiological surrogates than CBF.

Page 4 line 151: it would be quite an extreme interpretation of the hyperarousal model to assume that blood flow was increased globally, I would have thought a more reasonable interpretation would be an increase in areas related to arousal systems?

Page 4 line 157: as above for SPECT, PET is not only used to characterise metabolic rate/glucose metabolism, and FDG is not the only tracer.

Page 6 line 209: perhaps too minor to mention, but non-BOLD fMRI contrasts are available (e.g., CBF with ASL). Similarly on line 210, BOLD has volume and oxygenation contributions in addition to blood flow.

Page 7 line 234: it would help if this and other sub-section headings were underlined or similar.

Page 7 line 255: to emphasise this point, the work of Tagliazucchi and Laufs (Neuron 2014) makes the case that a large proportion of resting state scans contain sleep.

Page 8 line 286: the link between functional connectivity and arousal, in the sense of what would be expected in terms of regional and global functional connectivity changes with a change in arousal state, is not clear to me. It does not seem as straightforward to me to suggest that hyperarousal would mean higher functional connectivity.

Page 8 line 308: typo ‘of the pathophysiology’

Page 9 line 326: these regions would encompass the majority of the brain.

Page 9 line 339: do the authors have a view on the consistency of diagnostic categories of the patients including across some of the discrepant studies?

Page 9 line 347: the figure is not particularly clear, although I do not have any immediate suggestions for improving it.

Page 10 line 394-398: it would be worth being explicit about the method used to calculate functional or effective connectivity here (i.e., fMRI/EEG).

Page 11 line 412: does active wake mean task-engaged wake? The changes that are indicated (deactivation of DMN etc) would suggest so, but the phrase ‘active wake’ is ambiguous and could include the absence of stimuli, in which case the activations are not correct.

Page 11 line 432: wake drive is high in REM?

Page 11 line 447: an example of ‘cognitive training’ could be given.

Page 11 line 449: it could be more explicit in this paragraph that changes at in PI relative to GS.

Page 11 line 453: which high-density EEG studies?

Page 13 line 523: the study by Khalsa et al (Sleep 2016) would support the impact of sleep patterns on waking functional connectivity as well.

Author Response

General comments

The manuscript by Kay and Buysse reviews the insights provided by functional neuroimaging into the pathophysiology of insomnia, and applies a heuristic model of sleep-wake states to the specific case of insomnia. This is a very interesting topic, and an up to date review is needed. The manuscript itself it well written and generally clear, and makes a very good case for conceptualising insomnia as a complex disorder characterised by state- and region-dependent abnormalities in brain function. I have only a few relatively minor comments.

 Specific comments

Page 3 line 103: are these examples of higher GABA concentrations in good sleepers observed during wakefulness? Are there studies MRS studies during sleep in GS or PI?

The EEG state of participants is not typically assessed in MRS studies. The scans are relatively long and participants are allowed to stay awake or go to sleep. We note this as a potential limitation of prior MRS studies starting on line 84.

Page 3 line 110: in keeping with the general view expressed in the manuscript that PI is a complex, multifaceted disorder, I would be surprised if GABA was globally lower in PI compared to GS – would it not be more likely that there would be regionally-specific variations?

We agree and have added a clarifying statement to the end of the sentence to reflect that.

Page 3 line 122: typo, REM is an active sleep state

Thank you for catching that.

Page 3 line 128: source localization is a more common term than source generation.

              We agree and have made those changes.

Page 3 line 144: SPECT can be used with other tracers than 99mTc and other physiological surrogates than CBF.

We have rephrased our wording to reflect that.

Page 4 line 151: it would be quite an extreme interpretation of the hyperarousal model to assume that blood flow was increased globally, I would have thought a more reasonable interpretation would be an increase in areas related to arousal systems?

We agree and have added a clarifying statement to the end of the sentence to reflect that.

Page 4 line 157: as above for SPECT, PET is not only used to characterise metabolic rate/glucose metabolism, and FDG is not the only tracer.

We have rephrased our wording to reflect that.

Page 6 line 209: perhaps too minor to mention, but non-BOLD fMRI contrasts are available (e.g., CBF with ASL). Similarly on line 210, BOLD has volume and oxygenation contributions in addition to blood flow.

We have rephrased our wording to allow for other uses of fMRI and note that volume and oxygenation contribute to BOLD

Page 7 line 234: it would help if this and other sub-section headings were underlined or similar.

              We have underlined these sub-sections

Page 7 line 255: to emphasise this point, the work of Tagliazucchi and Laufs (Neuron 2014) makes the case that a large proportion of resting state scans contain sleep.

Thank you very much for sharing this reference. We have added it to support this point.

Page 8 line 286: the link between functional connectivity and arousal, in the sense of what would be expected in terms of regional and global functional connectivity changes with a change in arousal state, is not clear to me. It does not seem as straightforward to me to suggest that hyperarousal would mean higher functional connectivity.

We agree and have altered this sentence to reflect that point.

Page 8 line 308: typo ‘of the pathophysiology’

              We have corrected this error

Page 9 line 326: these regions would encompass the majority of the brain.

We take the reviewer’s point that those functions include very broad brain regions. We have re-written the statement to reflect that it is possible that insomnia involve altered functioning in several brain regions that play a role in those functions.

Page 9 line 339: do the authors have a view on the consistency of diagnostic categories of the patients including across some of the discrepant studies?

There is high consistency across studies for using DSM-IV criteria for PI. However, the criteria for PI are quite broad. Even within the category of DSM-IV PI, there is substantial variability among participants with regard to psychiatric/medical history and presenting symptoms (e.g., onset, maintenance, early morning awakenings).

Page 9 line 347: the figure is not particularly clear, although I do not have any immediate suggestions for improving it.

We have added clarification to the legend that we hope make the figure clearer.

Page 10 line 394-398: it would be worth being explicit about the method used to calculate functional or effective connectivity here (i.e., fMRI/EEG).

We now specify the method used and restrict our statement to the measure most relevant to this review (i.e., fMRI resting state functional connectivity).

Page 11 line 412: does active wake mean task-engaged wake? The changes that are indicated (deactivation of DMN etc) would suggest so, but the phrase ‘active wake’ is ambiguous and could include the absence of stimuli, in which case the activations are not correct.

              Yes and we added that clarification to the definition.

Page 11 line 432: wake drive is high in REM?

              REM sleep is a paradoxical state resembling both sleep and wake features. We propose here that the wake features of REM are driven by a rise in wake drive in the presence of high sleep drive. We altered our statement to indicate that there is heightened wake drive, not necessarily high wake drive during REM sleep.

Page 11 line 447: an example of ‘cognitive training’ could be given.

              We added additional details to provide more details about an example.

Page 11 line 449: it could be more explicit in this paragraph that changes at in PI relative to GS.

We altered the sentence to indicate that differences have been found in PI compared to GS.

Page 11 line 453: which high-density EEG studies?

              We added the references to the end of the sentence

Page 13 line 523: the study by Khalsa et al (Sleep 2016) would support the impact of sleep patterns on waking functional connectivity as well.

              We have added this supporting reference.

Reviewer 2 Report

This article reviews functional neuroimaging studies conducted in insomnia, and provides an original perspective on the interpretation of these findings. The conceptual framework proposed by the authors discusses possible mechanisms that extend beyond the classical ‘hyperarousal hypothesis’. In particular, the authors have pointed out the potential involvement of specific neural networks, such as the default-mode network, and the interplay between factors related to sleep drive, wake drive and self-perception of sleep-wake states (which they refer as conscious awareness). 

This review article offers a refreshing perspective on the pathophysiology of insomnia, and emphasizes the need for the identification of biomarkers and mechanisms underlying the complexity of insomnia, beyond the traditional hyperarousal concept.

I have a few minor comments

1.     Lines 234-236: please provide further details on the definition and functional significance of regional homogeneity and ALFF

2.     Line 266: typo, ‘decreased activity’ instead of ‘deceased activity’

3.     Heuristic model: the concept and terminology of ‘conscious awareness’ would benefit from a bit more clarification. Is it conceptualized here as the self-perception of sleep or wake states, the presence of mental content that can be retrieved upon awakening, etc.

4.     Conclusion: the authors mention the heterogeneity of insomnia; this concept might benefit from further explanation earlier in the manuscript and in relationship with the discrepancies between neuroimaging findings. How do the authors see this heterogeneity fitting in their model? For instance, how would sleep-state misperception vs insomnia with objectively short sleep duration be differentiated in this model?

Author Response

I have a few minor comments

1.     Lines 234-236: please provide further details on the definition and functional significance of regional homogeneity and ALFF

We have added content and references to help readers understand the definition and functional significance of these measures.

2.     Line 266: typo, ‘decreased activity’ instead of ‘deceased activity’

            We have corrected this error

3.     Heuristic model: the concept and terminology of ‘conscious awareness’ would benefit from a bit more clarification. Is it conceptualized here as the self-perception of sleep or wake states, the presence of mental content that can be retrieved upon awakening, etc.

            We have added a statement clarifying our conceptualization of this factor in the model.

4.     Conclusion: the authors mention the heterogeneity of insomnia; this concept might benefit from further explanation earlier in the manuscript and in relationship with the discrepancies between neuroimaging findings. How do the authors see this heterogeneity fitting in their model? For instance, how would sleep-state misperception vs insomnia with objectively short sleep duration be differentiated in this model?

            We have added several statements (including early in the manuscript) to further explain what is meant by heterogeneity of insomnia. We recognize that most of the neuroimaging studies were conducted on PI. We note that our model can be used to understand other subtypes and presentations of insomnia.  

Round 2

Reviewer 1 Report

The authors have responded comprehensively to my original comments, and I do not have any further issues to raise.